# DIG: Complex Layout Document Image Generation with Authentic-looking Text for Enhancing Layout Analysis

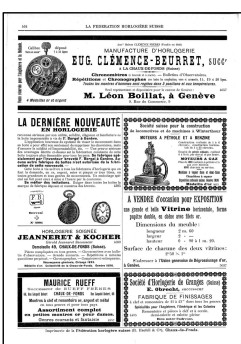 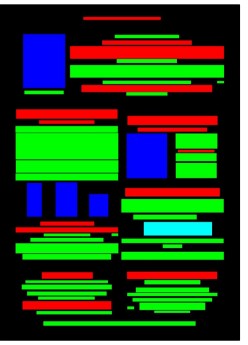 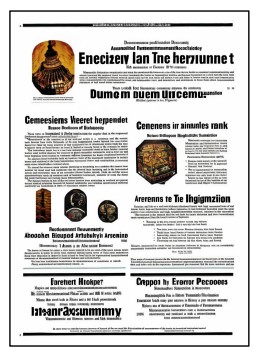 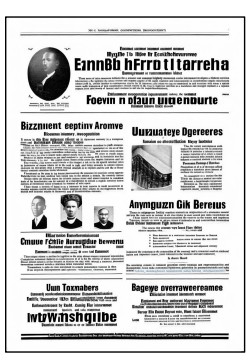 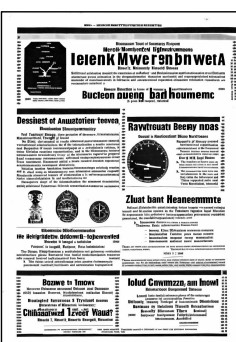

Real Image        Layout        Generated Images

**Figure 1: Generate corresponding document images based on the existing layout. One layout can generate an infinite number of diverse document images with complex layout and authentic-looking text.**

## ABSTRACT

Even though significant progress has been made in standardizing document layout analysis, complex layout documents like magazines and newspapers still present challenges. Models trained on standardized documents struggle with these complexities, and the high cost of annotating such documents limits dataset availability. To address this, we propose the Complex Layout **D**ocument **I**mage **G**eneration (DIG) model, which can generate diverse document images with complex layouts and authentic-looking text, aiding in layout analysis model training. Concretely, we first pre-train DIG on a large-scale document dataset with a text-sensitive loss function to address the issue of unreal generation of text regions. Then, we fine-tune it with a small number of documents with complex layouts to generate new images with the same layout. Additionally, we use a layout generation model to create new layouts, enhancing data diversity. Finally, we design a box-wise quality scoring function to filter out low-quality regions during layout analysis model training to enhance the effectiveness of using the generated images. Experimental results on the DSSE-200 and PRImA datasets show when incorporating generated images from DIG, the mAP of the layout analysis model is improved from 47.05 to 56.07 and from 53.80 to 62.26, respectively, which is a 19.17% and 15.72% enhancement compared to the baseline.

## CCS CONCEPTS

• **Applied computing** → **Multi / mixed media creation**; • **Computing methodologies** → *Object detection*.

## KEYWORDS

Complex layout document, Image generation, Document layout analysis, Authentic-looking text, Multimodal pre-training

## 1 INTRODUCTION

Document layout analysis aims to locate the component structures in document images [22], these include titles, text, images, tables, and other elements. Since it is usually the first step in all document analysis and understanding, the accuracy of this step directly affects all subsequent tasks [19].

The emergence of large-scale document layout analysis datasets [18, 23, 42] has advanced the performance of various deep learning models. However, these datasets are generally constructed through semi-automatic alignment of structured documents and document images. Although this provides sufficient data, it still has limitations: 1) The construction of these datasets requires structured documents like XML and LaTeX, primarily sourced from documents with standardized layouts like academic papers. This leads to the trained model failing to achieve satisfactory results on complex layouts beyond the dataset's scope. 2) The annotation granularity is fixed, which means manual data annotation is the only option for identifying unannotated or more fine-grained components. However, large-scale image annotation is inefficient and label-intensive. Therefore, exploring automated methods to acquire annotation images is crucial for enhancing complex layout analysis.

Recently, significant progress has been made in controllable image generation, enabling both text and image to control the process [26, 36, 40]. Therefore, a spontaneous idea is that **can we use layouts to control the generation of document images,**

**thus eliminating the labeling costs?** Recent works have utilized controllable image generation models to generate additional data for enhancing the image classification and semantic segmentation tasks. One of the common limitations of these works is that their target images are natural images. However, document images differ significantly from natural images in layout, content, and resolution requirements. Directly applying existing image generation models to generate document images leads to a huge gap from real images. Another key issue is the severe distortion in text regions, which make up a large portion of document images, rendering the generated images unsuitable for training layout analysis models.

To address these issues, we propose a general **d**ocument **i**mage **g**eneration (DIG) model. DIG is a two-step pre-training and controllable image generation model leveraging a large-scale document dataset. In the first step, we only train the autoencoder of the image generation model using a text-sensitive loss function, while keeping the control module frozen. This training strategy enables the model to generate images with authentic-looking text. In the second step, we freeze the image generation model and focus on training the control module. This allows the model to generate images based on the given layouts. After pre-training, we fine-tune DIG with complex layout documents to generate images with the same layouts. As shown in Figure 1, DIG can generate a variety of highly realistic document images with authentic-looking text according to one layout. To improve the diversity of layouts in generated images, we also train a layout generation model to generate new complex layouts as a supplement to DIG. DIG can generate document images that conform to the new layouts, as shown in Figure 2.

On a real scenario, the generated images may inevitably include some low-quality regions, which have negative effect on the training of layout analysis models. To tackle this issue, we design a box-wise quality scoring function to dynamically filter low-quality regions. Intuitively, low-quality regions will suffer higher losses, which inspires us to continuously update the average loss of every ground truth bounding box in generated images as its quality score. Bounding boxes higher than the score threshold are discarded, and their losses do not participate in model training.

Our contributions can be summarized as follows:

- We propose a new paradigm for obtaining complex layout document images by layout-controlled image generation. We train a document image generation model, DIG, using a text-sensitive loss function to ensure the generation of authentic-looking text. Using these generated images further enhances the performance of document layout analysis.
- We use a layout generation model to learn existing document layouts and generate new layouts. Then we use DIG to generate images corresponding to the new layout to improve the layout diversity.
- We design a box-wise quality scoring function that filters low-quality regions of the generated images during layout analysis model training, improving the efficiency of using the generated images.
- We apply the generated images by adding them to the training set or using them as a pre-training set. Experimental results demonstrate that generated images can significantly

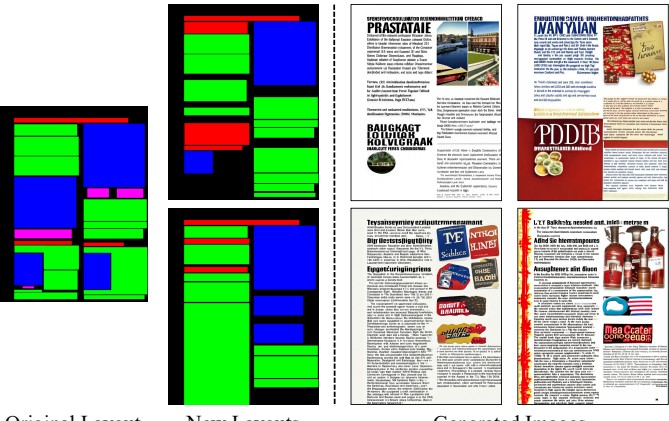

Original Layout    New Layouts    Generated Images

**Figure 2: Generate document images based on the new layout, which is generated from the original layout.**

improve the layout analysis model's capabilities on complex layout document images (mAP 47.05 → 56.07 on DSSE-200 and 53.80 → 62.26 on PRImA).

## 2 RELATED WORK

### 2.1 Document Layout Analysis Datasets

Document layout analysis datasets consist of document images and document components information. They are categorized into manually annotated and semi-automatic annotated.

Early datasets were mostly manually annotated and limited in quantity. SectLabel [20] includes 347 images from papers. PRImA [2] consists of 478 images from magazines and papers. DSSE-200 [38] contains 200 images, encompassing magazines, newspapers, slides, scanned documents, and papers. Recently, a large-scale manually annotated dataset, DocLayNet [23], has been introduced, consisting of 80,863 images from scientific, patent, manual, law, tender, and financial documents.

Before DocLayNet was proposed, large-scale datasets generated through semi-automatic annotation were widely used. PubLayNet [42] consists of over 360,000 images by matching papers in PDF and XML formats. Similarly, DocBank [18] is comprised of 500,000 images by matching papers in PDF and LaTeX formats. Models trained on these datasets performed well within their scopes but showed disparities when applied to documents from other domains.

To summarize from the above literature, there are three main problems with existing datasets: 1) Semi-automatic annotation requires structured documents, which typically have a simple layout. Therefore, models trained on such datasets are inadequate to handle complex layout analysis. 2) The high cost of manual annotation limits the scale of datasets. 3) Once the construction is completed, it is impossible to make any adjustments to the annotation granularity. Therefore, when applying the dataset to tasks with different annotation rules, reconstructing the entire dataset becomes unavoidable, representing the most pressing challenge within the existing dataset. To this end, a low-cost method for acquiring annotated images with complex layouts is necessary.

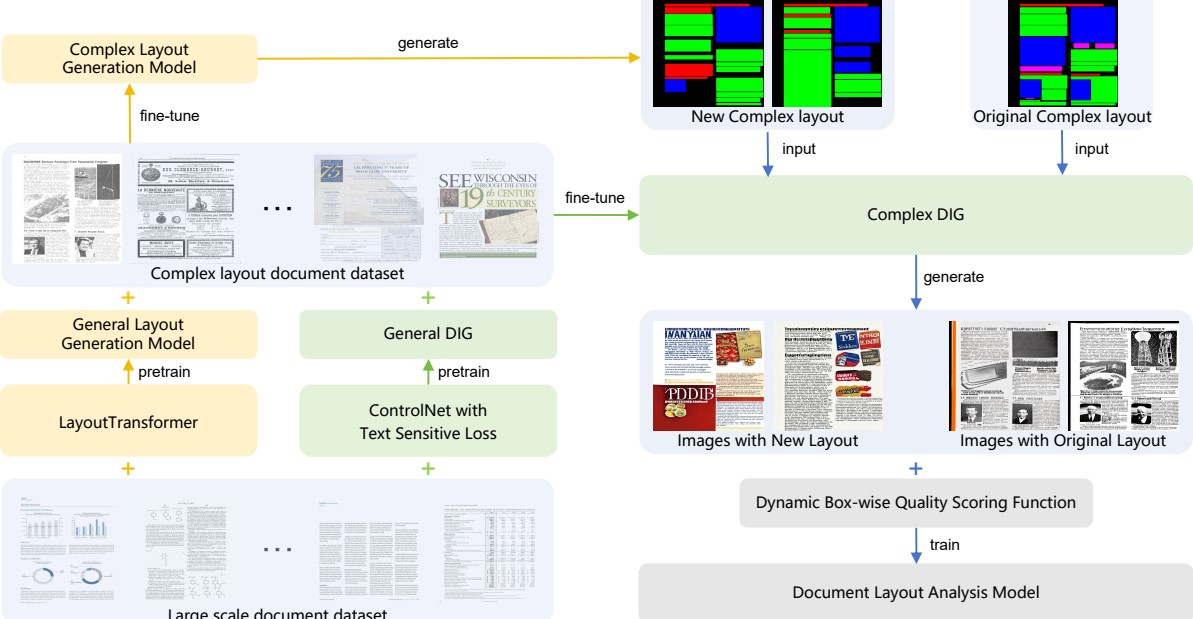

Figure 3: The pipeline of generating additional training data for document layout analysis through layout-controlled image generation. We pre-train DIG using a text-sensitive loss function for authentic-looking text generation. Under the control of original layouts and new layouts generated through a layout generation model, new images are generated. The quality scoring function which dynamically filters low-quality regions improves the effectiveness of utilizing the generated images.

## 2.2 Controllable Image Generation Model

The drawback of uncontrolled image generation models is that the input only consists of noise. As an improvement, controllable image generation models consider both text and images, which mainly involve three different structures: Generative Adversarial Networks (GANs) [8], Variational Autoencoders (VAE) [14], and Diffusion Models (DM) [11]. GANs consist of a generator and a discriminator, competing against each other to generate images. VAE encodes images into the mean and variance of a probability distribution, then generate latent space vectors through random sampling to reconstruct images. DMs gradually add noises to images and learn the reverse diffusion process through a U-net structure [27] to construct images from noises. Among them, DM-based methods [26, 35, 36, 40] have achieved the SOTA performance. For example, SDM [35] input the control conditions into the decoder of the U-net to guide image generation. Latent Diffusion Models (LDMs) [26] employ a pre-trained autoencoder to shift diffusion models from pixel to latent space, and cross-attention layers integrated into the U-net enable controllable image generation. FreestyleNet [36] applies Stable Diffusion (SD) [1] (LDMs trained on LAION-5B [29]) as the image generation model, and introduces a rectified cross-attention module to the U-Net to integrate control. ControlNet [40] preserves SD's parameters while incorporating a trainable U-Net encoder. This encoder connects to SD's U-Net decoder via zero convolutions. Control conditions are input into the trainable encoder and processed through zero convolutions to generate images, ensuring noise-free fine-tuning of SD.

## 2.3 Training Data Generation

Due to the impressive performance of controllable image generation models, some works have acquired additional training data by using GANs [3, 12, 15, 31, 41] or Diffusion Models [4, 10, 28, 32]. He et al. [10] and Azizi et al. [4] investigated the effectiveness of generated images for assisting image classification. Trabucco et al. [32] focused on using image generation models to edit existing training images. In the latest research, FreeMask [37] shared a similar motivation with us and they improved semantic segmentation performance using generated images.

Compared to tasks in the natural image segmentation domain like FreeMask, generating data for document layout analysis is more challenging. Image generation models commonly struggle with text rendering in images [7, 39], leading to a domain gap between generated "document images" and real images. Furthermore, compared to natural images, the visual differences between components of different categories are smaller in document images, while the visual differences between components of the same category are larger [6]. Hence, additional training of image generation models is necessary for generating document images.

## 2.4 Layout Generation Method

Generating images controlled by layouts can enhance the diversity of images with fixed layouts. However, when the number of existing layouts is limited, the diversity of layouts themselves becomes crucial. Therefore, we employ a layout generation model to generate layouts that conform to the patterns of existing ones, thereby enhancing the diversity of layouts in the generated images.

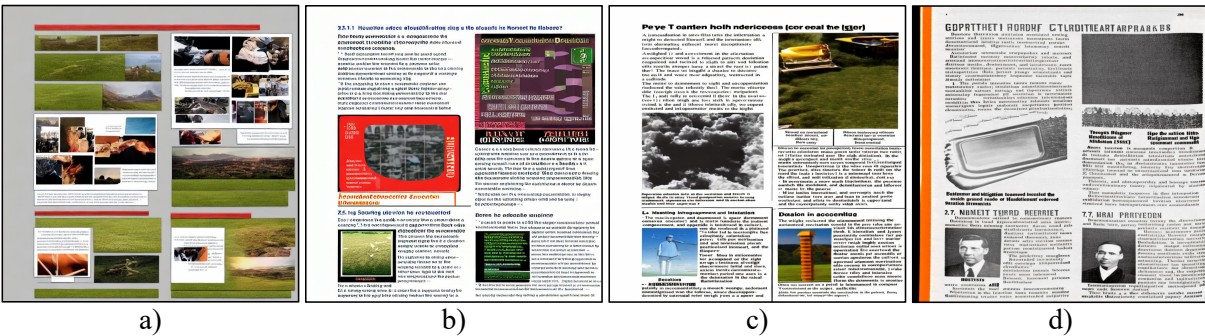

Figure 4: a) An image generated by ControlNet pre-trained on natural images. b) An image generated by ControlNet pre-trained on DocLayNet. c) An image generated by General DIG. d) An image generated by Complex DIG.

The goal of layout generation is to arrange a series of bounding boxes of different sizes and categories on an image under certain constraints. This task was first applied in the field of scene design, so many methods [17, 24, 33, 34] are proposed specifically for modeling room layouts. In scalable layout generation models, LayoutGAN [16] employed a GAN framework to generate semantic and geometric properties for a fixed number of scene elements. LayoutVAE [13] randomly generated image layouts given a set of labels. Manandhar et al. [21] utilized a graph network to develop an automatic encode framework for layout. LayoutTransformer [9] employed self-attention to construct an auto-regressive model that learned contextual relationships among layout elements, facilitating layout generation for a given domain. Its advantage lay in its ability to generate layouts from scratch or based on existing ones.

## 3 METHODOLOGY

### 3.1 Pipeline

Due to significant differences between document images and natural images, we pre-train a state-of-the-art controllable image generation model, ControlNet [40], using a large-scale document image dataset. Meanwhile, due to the lack of authenticity in text generation, we employ a text-sensitive loss function during pre-training. After pre-training, a general document image generation model (General DIG) is constructed. Then, we fine-tune General DIG on the a few complex layout documents to reduce the domain gap between simple layouts and complex layouts, resulting in Complex DIG. Subsequently, with just a few complex layouts, an infinite number of document images conforming to those layouts can be generated. Furthermore, we address the issue of insufficient diversity of layouts by incorporating a layout generation model. Finally, during the training of the layout analysis model, we utilize a box-wise quality scoring function that dynamically filters low-quality generated regions to enhance the effectiveness of utilizing the generated images. Figure 3 illustrates our pipeline.

### 3.2 Complex Layout Document Image Generation Pre-training

Though image generation models achieved tremendous success in natural images, applying ControlNet directly for generating document images yields poor quality, as illustrated in Figure 4a). This discrepancy stems from the fact that image generation models are primarily trained on natural images, whereas document images exhibit distinct layouts and structures like abundant text, lines, and tables. Considering the comprehensive coverage of document types and annotation categories in DocLayNet [23], we follow the training approach of ControlNet, freeze the parameters of Stable Diffusion and employ DocLayNet to pre-train ControlNet. This step helps mitigate overfitting that may arise from directly training ControlNet on a few complex layout documents. Layout images and text prompts used in pre-training are constructed using annotation files. The form of a text prompt is: "a document image of a [Document Category], including [number] [element], [number] [element], [number] [element],...". After pre-training ControlNet with DocLayNet, the generated images are more likely to have a correct layout, as shown in Figure 4b).

### 3.3 Authentic-Looking Text Rendering

To preserve the capabilities acquired from training on large-scale images, all parameters of Stable Diffusion within ControlNet are frozen, including the autoencoder used to decode latent space features back into pixel space. However, this autoencoder is trained on natural images and is not suitable for decoding outputs for text-rich document images. Therefore, after pre-training on DocLayNet, the generated images suffer from severe distortion in text regions.

To enhance the ability of Stable Diffusion to generate authentic-looking text, we conduct a two-step pre-training process by pre-training the autoencoder of Stable Diffusion before the pre-train step described in Section 3.2. In prior work, OCR-VQGAN [25] utilized a pre-trained text detector, CRAFT [5], as a text feature extractor to reconstruct figures rich in text. Inspired by this idea, we pass document images through the encoder $\mathcal{E}$ of autoencoder, obtaining latent space representations. These representations bypass the U-Net and are directly input into the decoder $\mathcal{D}$ of autoencoder to reconstruct images. Then, both the original and reconstructed images are fed into the pre-trained VGG [30] and CRAFT models to extract their perceptual features and text perceptual features, respectively. Perceptual features are obtained by taking the weighted average of output features from every convolutional layers in VGG. Then, we obtain text perceptual features from CRAFT the same way, since it also uses VGG as its backbone. By combining the reconstruction loss, perceptual loss, and text perceptual loss, we train

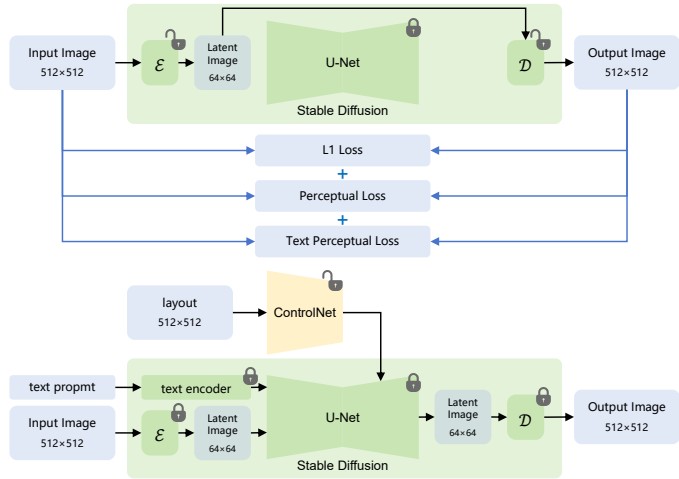

Figure 5: Train a general document image generation model through two-step pre-training: a) Pre-train the autoencoder of Stable Diffusion using a text-sensitive loss function while freezing other parameters, enabling the model to generate authentic-looking text regions. b) Pre-train ControlNet while freezing other parameters to generate document images with correct layouts.

the autoencoder until texts in the reconstructed images become authentic-looking:

$$\mathcal{L}_{rec} = ||Rec - Org||^2$$

$$\mathcal{L}_{perceptual} = \sum_l \frac{1}{I} \sum_i |VGG_l(Rec)_i - VGG_l(Org)_i||$$

$$\mathcal{L}_{\text{text-perceptual}} = \sum_l \frac{1}{I} \sum_i ||CRAFT_l(Rec)_i - CRAFT_l(Org)_i||$$

$$\mathcal{L} = \mathcal{L}_{rec} + \alpha \mathcal{L}_{perceptual} + \beta \mathcal{L}_{\text{text-perceptual}}$$

$$(1)$$

where $Org$ is the original image, $Rec$ is the reconstructed image. $VGG_l(Org)_i$ and $VGG_l(Rec)_i$ are the values of point $i$ of the feature map output by the convolutional layer $l$ in VGG of the original and reconstructed images, respectively. $CRAFT_l(Org)_i$ and $CRAFT_l(Rec)_i$ are defined analogously. $\alpha$ and $\beta$ are the weights of perceptual loss and text perceptual loss. The complete two-step pre-training for ControlNet is shown in Figure 5.

After the two steps of pre-training, General DIG is constructed. The generated images are very close to real document images in terms of layout and text authenticity, as shown in Figure 4c). However, a domain gap is still remaining compared to complex layout documents. So we fine-tune General DIG on a few complex layout documents while keeping Stable Diffusion fixed, resulting in Complex DIG. Images generated by it are shown in Figure 4d).

## 3.4 Document Layout Diversification

Utilizing existing layouts to generate images significantly increases the amount of trainable data. However, given the scarcity of samples in most complex layout document datasets, layout diversity

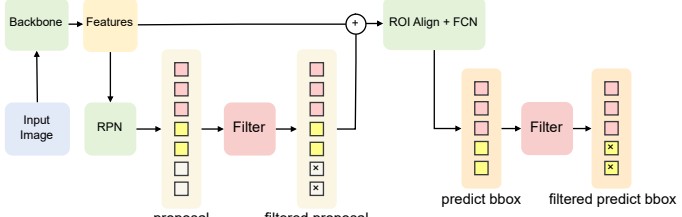

Figure 6: The pipeline of filtering low-quality regions using the box-wise quality scoring function during layout analysis model training. After RPN and FCN, Each bounding box is assigned some proposals based on IoU. Then we calculate the average loss for the assigned proposals as the quality score for that bounding box. Bounding boxes with scores higher than the dynamically changing average threshold are filtered out. In this figure, proposals assigned to the same bounding box have the same color.

is limited, potentially leading to overfitting during training. To address the overfitting issue, we employ a layout generation model to generate more diverse layouts. Nonetheless, generated layouts must conform to the patterns of real documents and cannot be entirely random. Therefore, we preserve a certain number of components from accurate layouts and utilize a layout generation model to complete them, thereby robustly increasing the diversity of layouts. We select LayoutTransformer [9] to fulfill this requirement.

LayoutTransformer discretizes coordinates in the image and models them using a categorical distribution to represent the probability of each discrete value. Each layout component is represented as $(s_i, x_i, y_i, w_i, h_i)$, where $s_i$ is the category, $(x_i, y_i)$ is the coordinate of the center point, $(w_i, h_i)$ is the size. Connect $n$ components into a sequence and embed start and end markers to form a sequence representation of length $5n + 2$. Use $\theta_j, j \in \{1, \ldots, 5n + 2\}$ to represent any element in the sequence, and use chain rules to model the joint distribution of all components in the layout as:

$$p(\theta_{1:5n+2}) = \prod_{j=1}^{5n+2} p(\theta_j | \theta_{1:j-1}) \tag{2}$$

We pre-train the model using layouts from DocLayNet and fine-tune it using a few complex layout documents. After generating a layout, we simultaneously save the layout image and bounding box information. The former serves as the control condition for DIG, while the latter serves as the ground truth during the training of the layout analysis model. Sometimes, overlapping components may occur in a generated layout, resulting in mismatches between the images generated from the layout and the ground truth. Therefore, post-processing of the generated layout is necessary. We sort the layout components in descending order based on their size and remove smaller components covered by larger ones. Figure 2 illustrates new layouts generated based on a portion of the original layout components.

Figure 7: Low-quality regions filtered by the box-wise quality scoring function are represented by different colored boxes: red for titles, blue for figures, magenta for captions, cyan for lists, and green for text.

## 3.5 Box-wise Quality Scoring Function for Dynamic Filtering of Low-Quality Regions

Inevitably, some low-quality regions are in the generated images, and we aim to minimize their negative impact. If the layout analysis model's prediction of a bounding box is highly inaccurate for real images, it indicates a hard case. However, this often suggests inadequate quality in the generated region for generated images. Motivated by this, we propose a box-wise quality scoring function for dynamically identifying and filtering such regions while training a layout analysis model based on object detection.

The typical approach of object detection generates anchors on feature maps obtained by a backbone, and uses a region proposal network (RPN) for binary classification and position regression on anchors, resulting in proposals. RPN loss is computed by comparing proposals with ground truths. Then, two fully connected networks (FCN) are used to classify proposals and refine their coordinates, and FCN loss is calculated by comparing these refined proposals with ground truths. The model is trained by optimizing both RPN loss and FCN loss; we call it the proposal-wise loss function.

Our box-wise quality scoring function evaluates the quality of each ground truth bounding box. Concretely, assume there are $N$ ground truth bounding boxes and $M$ proposals output by RPN or FCN. For the $n$-th bounding box $gt\_box_n$, we assign $K$ proposals above an IoU threshold to it thusly:

$$K = \sum_M \mathcal{T}\left(IoU(proposal\_box_m, gt\_box_n) > thres_{IoU}\right) \quad (3)$$

where $proposal\_box_m$ is the $m$-th proposal, $\mathcal{T}(x) = 1$ if $x$ is True, otherwise 0. Then we calculate the average RPN/FCN loss of $K$ proposals as the quality score of $gt\_box_n$:

$$\mathcal{S}_{box-wise_n} = \frac{1}{K}\sum_K \mathcal{F}(proposal\_box_k, gt\_box_n) \quad (4)$$

where $\mathcal{F}$ represents the RPN/FCN loss function. Next, we continuously update the mean quality score of all ground truth bounding boxes as a threshold:

$$\mathcal{S}_{box-wise} = \frac{1}{N}\sum_N \mathcal{S}_{box-wise_n} \quad (5)$$

If $\mathcal{S}_{box-wise_n} > \lambda \mathcal{S}_{bbox-wise}$, then $gt\_box_n$ is identified as a low-quality region and filtered. Consequently, the proposals assigned to it are ignored in the calculation of the proposal-wise loss. Then the proposals generated by RPN are no longer forwarded to

the subsequent FCN. Additionally, we set $\lambda$ to a relatively large value because smaller values of $\lambda$ might mistakenly filter some hard cases that could contribute to model training.

Throughout the training process, $N$ and $M$ accumulate continuously, causing the average box-wise quality score to change smoothly. This dynamic filtering mechanism gradually and accurately filters low-quality regions during training, allowing the model to focus more on learning correct layout features, thereby enhancing the efficiency of using generated images for layout analysis model training. Figure 6 shows the pipeline of filtering low-quality regions using the box-wise quality scoring function, and Figure 7 shows the filtered regions during training.

## 3.6 How to use the generated data

We utilize the generated images in two ways: joint training on both generated and real images and pre-training on generated images, followed by fine-tuning on real images. The former exposes the model to a broader data distribution, enhancing its generalization. The latter improves generalization while minimizing the impact of noise from generated images. During experimentation, we observed that images generated by original layouts performed better in joint training, whereas those generated by new layouts showed improved effectiveness in pre-training, as detailed in Section 4.2.

## 4 EXPERIMENT

### 4.1 Implement Details

Document layout analysis is mainly accomplished using instance segmentation or object detection. Considering that segmentation-based methods require additional post-processing steps to obtain a bounding box, we opt for object detection-based methods, whose results can be more easily applied to subsequent tasks. We select the widely used Faster R-CNN as an object detector, and do not use feature extractors pre-trained on a large-scale document dataset to eliminate the influence of other variables.

We utilize DocLayNet for pre-training ControlNet and Layout-Transformer, and choose DSSE-200 and PRImA as our target complex layout document datasets. For both datasets, we randomly select 100 images as the test set, the remaining images are used as the training set for fine-tuning ControlNet and LayoutTransformer, as well as training the layout analysis model. We use ControlNet to control the standard Stable Diffusion 1.5 model [1]. During the pre-training of ControlNet, we set the values of $\alpha$ and $\beta$ in Equation (1) to 1. After completing the pre-training and fine-tuning

**Table 1: The layout analysis performance of models trained solely on the generated images.**

| Dataset | Training data | mAP↑ | Dataset | Training data | mAP↑ |
|---|---|---|---|---|---|
| DSSE-200 | Real | 47.05 | PRImA | Real | 53.80 |
| | syn(1) | 33.97 | | syn(1) | 43.79 |
| | syn(5) | 41.42 | | syn(5) | 53.81 |
| | syn(10) | **45.72** | | syn(10) | **55.81** |
| | syn_layout(1) | 24.43 | | syn_layout(1) | 31.31 |
| | syn_layout(5) | 33.46 | | syn_layout(5) | 32.20 |
| | syn_layout(10) | 36.72 | | syn_layout(10) | 32.91 |

**Table 2: The layout analysis performance of joint training.**

| Dataset | Training data | mAP↑ | Δ | Dataset | Training data | mAP↑ | Δ |
|---|---|---|---|---|---|---|---|
| DSSE-200 | Real | 47.05 | | PRImA | Real | 53.80 | |
| | Real+syn(1) | 52.52 | ↑5.47 | | Real+syn(1) | 57.96 | ↑4.16 |
| | Real+syn(10) | **55.23** | **↑8.18** | | Real+syn(10) | **59.50** | **↑5.70** |
| | Real+syn_layout(1) | 51.66 | ↑4.61 | | Real+syn_layout(1) | 55.43 | ↑1.63 |
| | Real+syn_layout(10) | 53.82 | ↑6.77 | | Real+syn_layout(10) | 56.25 | ↑2.45 |

**Table 3: The layout analysis performance of models pre-trained on the generated images and fine-tuned on the real training set.**

| Dataset | Training data | mAP↑ | Δ | Dataset | Training data | mAP↑ | Δ |
|---|---|---|---|---|---|---|---|
| DSSE-200 | Real | 47.05 | | PRImA | Real | 53.80 | |
| | syn(10)→Real | 54.27 | ↑7.22 | | syn(10)→Real | 60.78 | ↑6.98 |
| | syn_layout(10)→Real | 54.27 | ↑7.22 | | syn_layout(10)→Real | 61.31 | ↑7.51 |
| | syn(10)→Real+syn_layout(10) | 53.44 | ↑6.39 | | syn(10)→Real+syn_layout(10) | 59.69 | ↑5.89 |
| | syn_layout(10)→Real+syn(10) | **56.07** | **↑9.02** | | syn_layout(10)→Real+syn(10) | **62.26** | **↑8.46** |
| | syn(10)→Real+syn(10) | 53.08 | ↑6.03 | | syn(10)→Real+syn(10) | 60.84 | ↑7.04 |

**Table 4: The performance of using images generated by a model that has not undergone pre-training with the text-sensitive loss function.**

| Training Data | mAP↑ |
|---|---|
| syn_without_text(10) | 43.71 |
| Real + syn_without_text(10) | 50.46 |
| syn_without_text(10)→Real | 51.93 |

**Table 5: FID scores between datasets generated by different models and the real dataset.**

| Generated dataset | fid↓ |
|---|---|
| syn(10) | 43.71 |
| syn_layout(10) | **40.86** |
| syn_without_text(10) | 75.19 |
| syn_without_ft(10) | 48.37 |

of ControlNet and LayoutTransformer, we generate 10 images for each layout in the training set, denoted as *syn* (10). The number in parentheses represents the multiple of data used relative to the real data. Additionally, we generate 5 new layouts for each layout and then generate 2 images for each new layout, denoted as *syn_layout* (10). During the generation of new layouts, we follow the rule of preserving 1 to 5 components from the accurate layout randomly. When filtering low-quality regions, we set $\lambda$ to 10 for DSSE-200 and 15 for PRImA. All training processes are conducted for 10,000 iterations. The metric used to evaluate the accuracy of layout analysis is mAP (mean Average Precision).

## 4.2 Enhancing Layout Analysis through Generated Images

First, we train Faster R-CNN solely using the training set of DSSE-200 and PRImA as baselines. Next, we train the model solely using generated images, employing various amount of *syn* and *syn_layout* as the training set. According to the results shown in Table 1, the performance improves as the number of generated images used increases, whether for *syn* or *syn_layout*. When using an equal number of the generated images, the result of *syn* are all higher than *syn_layout*, indicating that while *syn_layout* introduces layout diversity, it also introduces some noise. The best result obtained

solely from the generated images is achieved by *syn* (10). For DSSE-200, the mAP score is 45.72, which is only 1.33 lower than the baseline of 47.05. For PRImA, the mAP score reached 55.32, even surpassing the baseline of 53.80 by 2.01.

We further conduct joint training by incorporating the generated images into the real training set. According to the results shown in Table 2, all joint training results are significantly higher than the baseline. The best performance in joint training is achieved with DSSE-200+*syn* (10) and PRImA+*syn* (10), reaching 55.23 and 59.50, respectively, which is 8.18 and 5.70 higher than the baseline, demonstrating the effectiveness of joint training. In both scenarios of training solely with the generated data and joint training, the performance of *syn* surpasses that of *syn_layout*. Therefore, consider using *syn_layout* as pre-training data to enhance the generalization ability while reducing the impact of noise.

Finally, we utilize the generated images for pre-training. As indicated in Table 3, for DSSE-200, the results of pre-training with *syn* (10) and *syn_layout* (10) are both 54.27. For PRImA, models pretrained by *syn* (10) and *syn_layout* (10) reach 60.78 and 59.17, respectively. Compared to the baseline, all models have evidently boosted by larger than 6. Considering that both joint training and pre-training can enhance model performance, we combine the two ways. Specifically, we use one of *syn* and *syn_layout* for pre-training, and the other for joint training, namely *syn* (10)→ Real+*syn_layout* (10) and *syn_layout* (10)→Real+*syn* (10). The former performs similarly compared to solely conducting joint training or pre-training. However, the latter achieves the best results among all combinations of data for both datasets, reaching 56.07 and 62.26, which is a 19.17% and 15.72% enhancement compared to the baseline. This confirms that *syn_layout* is better suited for pre-training, whereas *syn* is more suitable for joint training, as hypothesized. To further demonstrate that view, we also use *syn* for both pre-training and joint training as a comparison. The experiment showed a decrease in performance on both datasets.

## 4.3 Ablation Studies

To validate the effectiveness of employing the text-sensitive loss function, we opt not to pre-train the autoencoder of Stable Diffusion but instead directly perform the second step of pre-training. We use DSSE-200 as the target dataset. Based on the results shown in Table 4, whether for joint training or pre-training, the performance significantly decreases (55.23 → 50.46, 54.27 → 51.93). Nevertheless, the results are still higher than the baseline, proving the robustness of using the generated images to assist the layout analysis training.

To validate the effectiveness of box-wise quality scoring funtion and determine the most effective filtering threshold, we conduct experiments with the *syn_layout* (10)→Real + *syn* (10) setup, which yields the highest mAP score. We test values of $\lambda$ at 2, 5, 10, 15, 20 and 25, as well as the baseline without using a scoring function. According to the results shown in Figure 8, the mechanism for filtering low-quality regions demonstrates strong robustness, as the results under various values exceed the baseline. The best performance is observed when set $\lambda$ to 10 in DSSE-200 and 15 in PRImA.

Using generated images as training data to obtain the mAP metric for layout analysis models is the most direct way to evaluate the

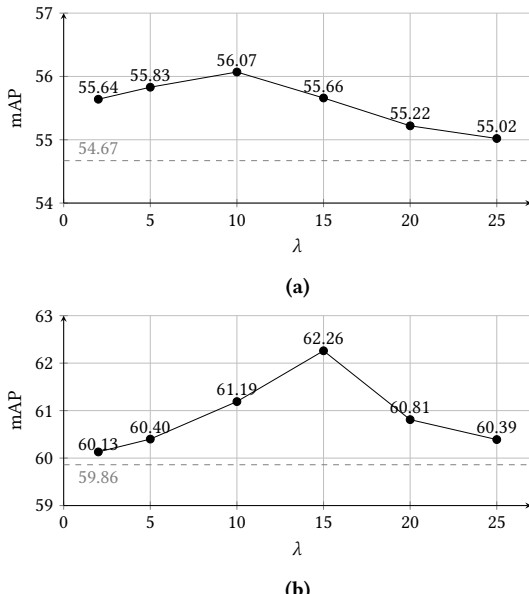

**Figure 8: The impact of different filter factors on (a) DSSE-200 dataset and (b) PRImA dataset.**

quality of generation, but FID score can still serve as a supplementary indicator of generation quality. We computed the FID scores for *syn* (10), *syn_layout* (10), *syn_without_text* (10) and *syn_without_ft* (10) with the real data from DSSE-200, where *syn_without_ft* (10) are generated by the General DIG without fine-tuning on DSSE-200 training set. As shown in Table 5, *syn_layout* (10) exhibit the best FID score, with *syn* (10) and *syn_without_ft* (10) slightly inferior to it. However, *syn_without_text* (10) falls far behind the other generated datasets, highlighting the importance of the text-sensitive loss function for generating document images.

## 5 CONCLUSION

We introduce a method of utilizing controllable image generation models to generate document images as additional training data, thereby enhancing the model's capability to analyze complex layout documents. Leveraging a text-sensitive loss function and a large-scale document dataset, we train a general document image generation model, DIG, capable of producing document images with authentic-looking text. Next, we leverage a layout generation model to enhance the diversity of document layouts in the generated dataset. When using generated images for training, we develop a box-wise quality scoring function to filter out low-quality regions in the generated images. This ensures the model prioritizes high-quality information during training, enhancing the effectiveness of utilizing the generated images. Experiments demonstrate significant enhancements in layout analysis performance, whether by using the generated images for pre-training or joint training with real images. Furthermore, images generated from new layouts are better suited for pre-training, while those generated from existing layouts are more appropriate for joint training.

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
