# OpenReview forum: "DIG: Complex Layout Document Image Generation with Authentic-looking Text for Enhancing Layout Analysis"
_acmmm.org/ACMMM/2024/Conference — MM2024 Poster_

### Official Review · Reviewer_yBMx · 2024-05-22

**Rating:** 4
**Confidence:** 3

**Summary:**

This paper presents Document Image Generation, which generates diverse document images with complex layouts and authentic-looking text for document layout analysis. Additionally, a box-wise quality scoring function is developed to filter low-quality regions of the generated images during layout analysis model training. Experiments demonstrate that the proposed model improves the performance of the layout analysis model.

**Strengths:**

The idea of generating training data for document layout analysis is interesting, and I believe it has practical application value. This paper develops a diffusion-based method for obtaining complex layout document images through layout-controlled image generation. To ensure the generation of authentic-looking text, the authors further introduce a text-sensitive loss function to the model. Experiments demonstrate the effectiveness of the proposed method.

**Limitations:**

1. Lack of experiments to verify the proposed method on popular benchmarks such as PubLayNet, DocBank, and DocLayNet.

2. Lack of experiments to compare to Controlnet. It would be nice to have quantitative results to compare with Controlnet.

3. It seems that the resolution of the generated image is 512*512. However, the resolution of document images is usually larger. Is there a way to increase the resolution of the generated image? If so, it would be nice to conduct an experiment on increasing the resolution of the generated image.

**Suitability:**

2

---

### Official Review · Reviewer_X4XV · 2024-05-22

**Rating:** 3
**Confidence:** 3

**Summary:**

The article addresses the poor performance of existing document layout analysis models in handling complex layout documents and the high cost of annotation. It proposes the DIG model, which generates document images with realistic text through a two-step pretraining and fine-tuning process. The model also introduces a layout generation model to increase the diversity of training data and a box-wise quality scoring function to filter out low-quality regions. Experimental results show that when combined with images generated by DIG, the mAP of the layout analysis model is significantly improved compared to the baseline model on the DSSE-200 and PRImA datasets.

**Strengths:**

- The proposed method significantly improves the performance of the layout analysis model in practical use. Among the two methods used in the article: joint training and pre-training on generated images, both can bring good improvements to the mAP index of the document layout analysis model.
- The explanation and elaboration of the results of various parameters are relatively clear, and the experimental demonstration is relatively sufficient.

**Limitations:**

1. Although the article uses a text-sensitive loss function, the generated image text is still unrecognizable and unrealistic. There is no visualization to compare the impact of this loss function.

2. There is no comparison with the results generated by other methods. Does FreeMask mentioned in related work have the potential to be adapted as a baseline for comparison?

3. Compared to ControlNet, DIG appears to have only added some loss functions.

**Suitability:**

3

---

### Official Review · Reviewer_uYL7 · 2024-05-27

**Rating:** 4
**Confidence:** 3

**Summary:**

This paper focused on improving layout analysis model performance for complex layout documents by training on generated layout documents. This paper proposed Complex Layout Document Image Generation (DIG) model to achiveve this purpose. Specifically, this paper
- 1. pre-train DIG on a large-scale document dataset with a text-sensitive loss function;
- 2. fine-tune DIG with a small number of documents with complex layouts to generate new images with the same layout;
- 3. use a layout generation model to create new layouts, enhancing data diversity;
- 4. design a box-wise quality scoring function to filter out low-quality regions during layout analysis model training.
Results show that the training on the synthetic dataset generated by DIG improves the performance of layout analysis model.

**Strengths:**

1. With the rapid development of generative models like stable diffusion, the motivation of using synthetic dataset to improve the discriminative tasks is meaningful and useful. The overall constrcution pipeline of synthetic training dataset is solid.
2. The improvement on object detection task is significant, where a 19.17% and 15.72% improvement are achieved, which in turn validate the effectiveness of the motivation

**Limitations:**

1. The main concern lies in the inadequate downstream tasks and baseline methods. This paper only validates the effectiveness of training on synthetic dataset on object detection tasks, and only Faster R-CNN is used as the layout analysis model, which is quite old. More downstream documents analysis tasks and newly baseline documents analysis models should be included.
2. Based on weakness 1, is the large improvement simply based on the too poor performance of Faster R-CNN? Since it is a very old method. I wonder whether the improvement brought by the proposed DIG will significanly decrease when switch to new state-of-the-art layout documents analysis model
3. The quality of generated text is still poor, and are usually non-sense sequence. Will this decrease other documents analysis tasks like OCR?

**Suitability:**

2

---

### Meta-Review · Area_Chair_6EQH · 2024-07-01

**Recommendation:** Accept (Poster)
**Confidence:** 5

**Metareview:**

This paper proposes a new method for obtaining complex layout document images through layout-controlled image generation. The main motivation for using synthetic datasets to improve discriminative tasks is meaningful, especially with the rapid development of generative models like stable diffusion. And the overall construction pipeline of the synthetic training dataset is effective. Specifically, the improvement in the object detection task is significant, with 19.17% and 15.72% improvements achieved.

Therefore, despite mixed ratings (2 BA, 1 BR), I tend to accept this paper for its practical value and fresh perspective on the document analysis community. The authors should further revise the paper according to the reviewers' suggestions.